# Status of Omics Research Capacity on Oral Cancer in Africa: A Systematic Scoping Review Protocol

Lawrence Achilles Nnyanzi [1,2,†], Akinyele Olumuyiwa Adisa [3,†], Kehinde Kazeem Kanmodi [1,4,5,*,†], Timothy Olukunle Aladelusi [6,†], Afeez Abolarinwa Salami [4,6], Jimoh Amzat [7,8], Claudio Angione [9], Jacob Njideka Nwafor [4,10], Peace Uwambaye [11], Moses Okee [12], Shweta Yogesh Kuba [1], Brian Mujuni [12], Charles Ibingira [12], Kalu Ugwa Emmanuel Ogbureke [13] and Ruwan Duminda Jayasinghe [14,15]

1   Head and Neck Cancer Working Group, School of Health and Life Sciences, Teesside University, Middlesbrough TS1 3BX, UK
2   School of Public Health, King Ceasar University, Kampala P.O. Box 88, Uganda
3   Department of Oral Pathology, University of Ibadan, Ibadan 200005, Nigeria
4   Campaign for Head and Neck Cancer Education (CHANCE) Programme, Cephas Health Research Initiative Inc., Ibadan 200005, Nigeria
5   Faculty of Dentistry, University of Puthisastra, Phnom Penh 120204, Cambodia
6   Department of Oral and Maxillofacial Surgery, University College Hospital/University of Ibadan, Ibadan 200005, Nigeria
7   Department of Sociology, Usmanu Danfodiyo University, Sokoto 84001, Nigeria
8   Department of Sociology, University of Johannesburg, Johannesburg 2001, South Africa
9   School of Computing, Engineering, and Digital Technologies, Teesside University, Middlesbrough TS1 3BX, UK
10  Division of Medicine, Nottingham University Teaching Hospital NHS Trust, Nottingham NG7 2GT, UK
11  Department of Preventive and Community Dentistry, University of Rwanda, Kigali 4285, Rwanda
12  College of Health Sciences, Makerere University, Kampala P.O. Box 88, Uganda
13  School of Dentistry, University of Texas Health Science Center at Houston, Houston, TX 77054, USA
14  Department of Oral Medicine and Periodontology, Faculty of Dental Sciences, University of Peradeniya, Peradeniya 20400, Sri Lanka
15  Center for Research in Oral Cancer, Faculty of Dental Sciences, University of Peradeniya, Peradeniya 20400, Sri Lanka
*   Correspondence: kkanmodi@puthisastra.edu.kh or k.kanmodi@tees.ac.uk or kanmodikehinde@yahoo.com
†   These authors contributed equally to this work.

**Abstract:** Over the past decade, omics technologies such as genomics, epigenomics, transcriptomics, proteomics, and metabolomics have been used in the scientific understanding of diseases. While omics technologies have provided a useful tool for the diagnosis and treatment of diseases globally, there is a dearth of literature on the use of these technologies in Africa, particularly in the diagnosis and treatment of oral cancer. This systematic scoping review aims to present the status of the omics research capacity on oral cancer in Africa. The guidelines by the Joanna Brigg's Institute for conducting systematic scoping reviews will be adopted for this review's methodology and it will be reported using the Preferred Reporting Items for Systematic Reviews and Meta-Analyses extension for Scoping Reviews (PRISMA-ScR) checklist. The literature that will be reviewed will be scooped out from PubMed, SCOPUS, Dentistry and Oral Sciences Source, AMED, CINAHL, and PsycInfo databases. In conclusion, the findings that will be obtained from this review will aid the in-depth understanding of the status of oral cancer omics research in Africa, as this knowledge is paramount for the enhancement of strategies required for capacity development and the prioritization of resources in the fight against oral cancer in Africa.

**Keywords:** omics; oral cancer; research; capacity; Africa; scoping review

## 1. Introduction

Science has evolved from looking through cellular biology to omics of diseases. Omics focuses on the combined description of biological molecules that account for the structure, function, and specifics of an organism [1]. The main principle driving omics methods is that a complicated organism can be understood better if studied as a whole [2].

"Omics" sciences include transcriptomics, genomics, metabolomics, proteomics, metagenomics, and epigenomics [3]. Transcriptomics encompasses everything relating to RNAs. This includes their transcription and expression levels, functions, locations, trafficking, and degradation [4]. Transcriptomics covers all types of transcripts, including messenger RNAs, microRNAs, and different types of long noncoding RNAs [4]. It also includes the structures of transcripts and their parent genes with regards to start sites, 5′ and 3′ end sequences, splicing patterns, and posttranscriptional modifications [4]. Genomics focuses on the structure, function, evolution, mapping, and editing of an organism's complete set of DNA, including all of its genes as well as its hierarchical, three-dimensional structural configuration [5]. Metabolomics is the comprehensive analysis of metabolites in a biological specimen [6]. The technologies used for this transcend the scope of standard clinical chemistry techniques and are capable of precise analyses of thousands of metabolites and can thus establish the metabolic phenotypes of a sample [6]. Proteomics enables us to identify proteins, study their structure, know their function, and map their interactions (including protein–protein interactions) in a cellular context [7]. Metagenomics involves genomic analysis of microorganisms by direct extraction and cloning of DNA from their natural environment [8,9]. Unlike traditional single-genomics approaches, metagenomics does not rely on having to singularize individual bacterial clones from complex microbial mixtures, but catalogs by sequencing all genes and genomes from a mixed community at once [9]. Epigenomics is the study of all of the epigenetic changes in a cell [10]. These are changes in the way genes are switched on and off without changing the actual DNA sequence [10].

Generally, omics-based research output in the global literature has been from countries outside Africa, and although more African-based omics research is being seen in the literature, the continent is still trailing behind in the development and wide use of this scientific method. However, such studies are crucial as genetic makeup and tumor biology varies in different populations. The potential for genomics research in Africa is comparatively low and this has hindered optimal benefits from genomics applications in medicine and clinical practice. It is now clear that the omic layers do not act in isolation [11]. Conversely, their complex interplay is a key factor in several diseases, and directly informs the observable disease phenotype. Therefore, multi-omic approaches and a systems-level view are paramount to fully understanding a disease phenotype [3–11].

A recent scoping review that evaluated cancer-related omics research between 2012 and 2019 from the African continent focused on publications on prostate cancer, colorectal cancer, ovarian cancer, hepatocellular carcinoma, endemic Burkitt's lymphoma, and esophageal squamous cell carcinoma [3]. However, omics research on oral cancer (oral squamous cell carcinoma) was not included in the review [3]. This suggests two things: firstly, there is limited capacity for omics research on oral cancer in Africa and, secondly, omics research on oral cancer is a neglected research area in Africa.

There are currently significant differences in genomics research capability among African countries, with South Africa having the highest research performance in genomics [3]. This is because South Africa has made significant investments in building its genomics and biotechnology program. The main challenges limiting the development of omics approach to research in most of African countries are lack of or insufficient basic infrastructure, ill-equipped laboratories, lack of expertise, inadequate connectivity to research centers, and lack of training programs in bioinformatics and omics strategies [3]. These challenges explain why cancer omics is poorly explored in Africa [12].

It has been projected that oral cancer cases in Africa will keep increasing [13]. As projected, oral cancer cases will reach approximately 29,583 in the year 2020, 37,715 in 2030,

and 57,327 in 2050 [13]. This shows a significant progressive increase in the number of such cases and thus advanced research is needed to understand the biology of oral cancer and to develop therapeutic interventions that are more effective in curing the disease. It is however pertinent that the status of oral cancer omics research capacity in Africa be evaluated.

However, after a scoping search of notable databases—PubMed, SCOPUS, Web of Science, CINAHL Ultimate, and APA PscyInfo—for studies evaluating the status of oral cancer omics research capacity in Africa, no known scoping review on such topic area was found. The availability of a scoping review evidence on this area is very crucial for the in-depth and contemporary understanding of this research landscape on the continent, as such evidence will set the pace for the growth and development of oral cancer omics research capacity in the African scientific community.

To fill this current void of evidence, the Consortium for Head and Neck Cancer in Africa, formerly called the International Head and Neck Cancer Working Group [IHNCWG], seeks to conduct such review [14]. Hence, this paper proposes a systematic scoping review that aims to critically evaluate the status of omics research capacity on oral cancer in Africa.

## 2. Methods

### 2.1. Review Design

The design of this study will be based on the guidelines of the Joanna Brigg's Institute for conducting systematic scoping reviews [15], and the study will be reported based on the Preferred Reporting Items for Systematic Reviews and Meta-Analyses extension for Scoping Reviews checklist (see Appendix A [Table A1]) [16]. In addition, the quality of the methodological process of this scoping review will be informed by the Assessment of Multiple Systematic Reviews (AMSTAR-2) tool (see Appendix A [Table A2]) [17,18].

### 2.2. Review Question

This study seeks to address this principal question: "What is the status of omics research capacity in oral cancer in Africa?".

### 2.3. Literature Selection Criteria

The inclusion or exclusion of a literature into this scoping review will be informed by a group of criteria, which are listed below:

### 2.3.1. Inclusion Criteria

1.  All forms of peer-reviewed journal publications on oral cancer omics in which an African researcher (i.e., a researcher affiliated to an organisation in Africa) is an author/co-author.
2.  Publications published in the English language.
3.  Publications in which the full text is accessible.

### 2.3.2. Exclusion Criteria

1.  Publications on oral cancer omics in which an African researcher is not an author/co-author.
2.  Publications on omics in which an African researcher is an author/co-author, and that are not focused oral cancer.
3.  Publications that are not published in peer-reviewed journals.
4.  Publications with full texts that are inaccessible.
5.  Publications published in any language other non-English language.

### 2.4. Literature Search Strategy

The literature search will be based on the PCC (population [p], concept [c], and context [C]) framework [19]. In this proposed scoping review, the population in focus is researchers affiliated to African institutions, the concept is omics research, and the context is oral cancer. Search terms, as shown in Table 1, which are search terms and synonyms, will be used for the literature search. Without limiters, six research databases will be searched with the aid

of the identified search terms, Boolean operators ("AND" and "OR"), and truncations ("*" and "#") to retrieve relevant literature on digital interventions on OC: PubMed; SCOPUS; Dentistry and Oral Sciences Source; AMED—The Allied and Complementary Medicine Database; CINAHL Complete; and APA PsycInfo.

**Table 1.** Search Combination.

| PCC Framework | Focus | Scope of Database Search | Search Terms |
|---|---|---|---|
| Population | Researchers affiliated to institutions in African countries, territories, and dependencies | Affiliation search | "Algeria", "Angola", "Benin", "Botswana", "Burkina Faso", "Burundi", "Cape Verde", "Cabo Verde", "Cameroon", "Central African Republic", "Chad", "Comoros", "Congo", "Cote D'ivoire", "Ivory Coast", "Djibouti", "Democratic Republic of Congo", "Egypt", "Equatorial Guinea", "Eritrea", "Eswatini", "Ethiopia", "Gabon", "Gambia", "Ghana", "Guinea", "Guinea Bissau", "Kenya", "Lesotho", "Liberia", "Libya", "Madagascar", "Malawi", "Mali", "Mauritania", "Mauritius", "Morocco", "Mozambique", "Namibia", "Niger", "Nigeria", "Rwanda", "Sao Tome And Principe", "Senegal", "Seychelles", "Sierra Leone", "Somalia", "South Africa", "South Sudan", "Sudan", "Tanzania", "Togo", "Tunisia", "Uganda", "Zambia", "Zimbabwe", "Reunion", "Saint Helena", "Western Sahara", and "Mayotte" |
| Concept | Omics | All fields search | "omics", "proteomics", "metabolomics", "transcriptomics", "genomics", "sociogenomics", "metagenomics", "phenomics", "gene", and "genetics" |
| Context | Oral cancer | All fields search | "Oral cancer", "oropharyngeal cancer", "oral squamous cell carcinoma", "oral cavity cancer", and "cancer of the lip" |

### 2.5. Deduplication of Literature

The Rayyan software will be used to deduplicate the outputs retrieved from the literature search [20].

### 2.6. Literature Screening and Selection

With the aid of the Rayyan software [20], all deduplicated literature will be screened based on the established selection criteria. The screening process will be two-staged and at least three independent reviewers who were oral oncology researchers will be involved: two reviewers will screen all the deduplicated literature, while the third reviewer will resolve the conflicts in the screening decisions made by the other two reviewers in case there is any. Specifically, the first stage will involve title and abstract screening, while the second stage will involve full text screening. Only the literature that met the inclusion criteria will be included into the SR.

### 2.7. Quality Appraisal of the Included Literature

The included literature will be appraised for its quality using the Mixed Methods Appraisal Tool (MMAT) 2018 version (Table 2) [21].

MMAT grades an article on a scale of 0 to 7 using a set of seven questions, where the first two questions are general questions for all study designs, while the remaining five questions are study-design specific, covering qualitative study design, quantitative randomized control trial design, quantitative non-randomized design, quantitative descriptive design, and mixed methods design. The grading approach that was used in this proposed

scoping review was adopted from Clark, Chisnall, and Vindrola-Padros [22]. Hence, in the grading process, a response of "Yes" to an appraisal question will be scored 1 point, while a response of "No" or "I cannot tell" to an appraisal question will be scored 0 or 0.5 point, respectively. After all the seven appraisal questions were answered for each appraised article, and each answer has been given a score, these scores will be summed up to determine the level of quality for such an article. For each appraised article, a cumulative score range of 4 to 7 points will be rated as above average quality, a cumulative score of 3.5 points will be rated as average quality, and a score range of 1 to 3 points will be rated as below average quality. As the proposed study is a scoping review, all of the included articles will be reviewed, regardless of the quality appraisal outcome. The essence of the quality appraisal in this proposed scoping review was just to evaluate the scientific rigor of the existing studies conducted on oral cancer omics research by African researchers, not otherwise.

**Table 2.** Quality appraisal table format for the assessment of article(s) that will be included.

| No. | Author(s) (Year) | Study Design | MMAT Version 2018 Questions (Hong et al., 2018) * | | | | | | | Total Score (Over 7) | Grading | Status |
|---|---|---|---|---|---|---|---|---|---|---|---|---|
| | | | General Screening Questions | | Questions Specific to Study Design | | | | | | | |
| | | | S1 | S2 | 1st | 2nd | 3rd | 4th | 5th | | | |

S1—Screening question 1; S2—Screening question 2; * Details of the Mixed Methods Appraisal Tool version 2018, by Hong et al.'s can be accessed by downloading this document [21].

### 2.8. Data Extraction, Collation and Charting

Data will be extracted from the literature that were included in this SR via a customized data extraction form (Table 3). These data include citation data (names of authors and publication year), affiliation names of authors from African institutions, publication type, research design, research objectives, geographical location (country) of the study population (sample), study population (sample) characteristics, sample size, study instruments, findings, limitations, and conclusions. After the extraction of these data, data collation and summarization into themes will be done. The summarized data will be presented using texts, figures (e.g., Figure 1), and tables. Texts will be used to narrate the findings, while figures and tables will be used to summarize or caption the findings.

**Table 3.** Data extraction form.

| Author | African Institution | Publication Year | Research Design | Research Objective | Location of Study Population (Sample) | Study (Sample) Population Characteristics | Sample Size | Study Instruments | Findings | Limitations | Conclusions |
|---|---|---|---|---|---|---|---|---|---|---|---|
| | | | | | | | | | | | |

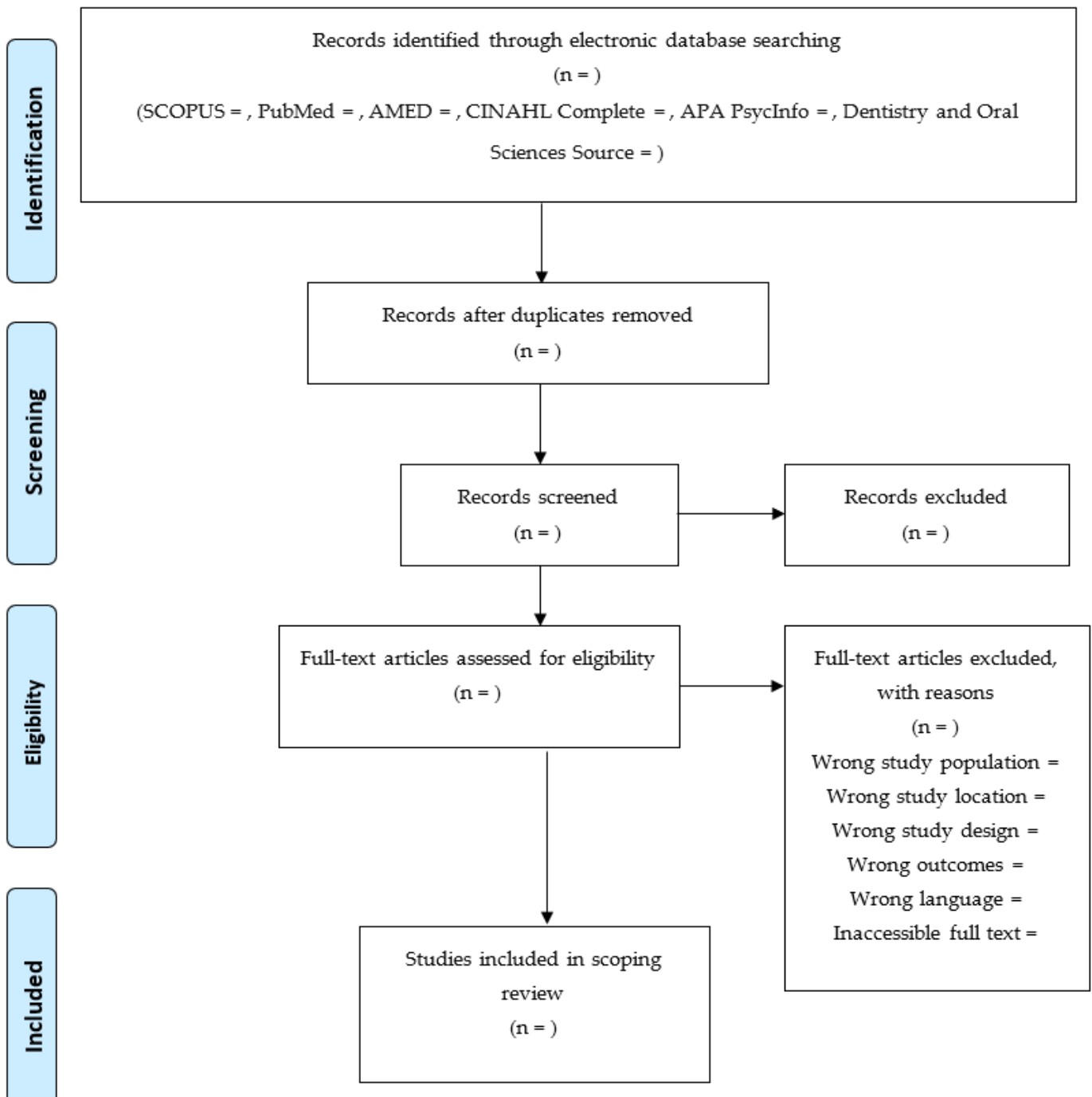

**Figure 1.** Flow chart of literature search and sorting process.

### 3. Conclusions

The results from this study will highlight the depth of status of omics research capacity in Africa, providing a unique opportunity to develop targeted capacity development approaches. This work will support the prioritization of resources in the areas that need more resourcing to enhance greater use of omics technologies in the diagnosis and management of oral cancer in Africa. Recommendations from this study could be scaled to other low- and middle-income countries with similar settings as those in Africa.

**Author Contributions:** Conceptualization, K.K.K., L.A.N., A.O.A., T.O.A., K.U.E.O. and R.D.J.; methodology, K.K.K.; software, K.K.K.; validation, K.K.K., L.A.N., A.O.A., T.O.A., A.A.S., J.A., C.A., J.N.N., P.U., M.O., S.Y.K., B.M., C.I., K.U.E.O. and R.D.J.; formal analysis, K.K.K. and A.A.S.; investigation, K.K.K., L.A.N., A.O.A., T.O.A., A.A.S., J.A., C.A., J.N.N., P.U., M.O., S.Y.K., B.M., C.I., K.U.E.O. and R.D.J.; resources, K.K.K., L.A.N., A.O.A., T.O.A., A.A.S., J.A., C.A., J.N.N., P.U., M.O., S.Y.K., B.M., C.I., K.U.E.O. and R.D.J.; data curation, A.O.A., K.K.K., T.O.A. and L.A.N.; writing—original draft preparation, A.O.A., K.K.K., T.O.A. and L.A.N.; writing—review and editing, K.K.K., L.A.N., A.O.A., T.O.A., A.A.S., J.A., C.A., J.N.N., P.U., M.O., S.Y.K., B.M., C.I., K.U.E.O. and R.D.J.; visualization, K.K.K.; supervision, K.U.E.O. and R.D.J.; project administration, K.K.K.; funding acquisition, L.A.N., A.O.A., T.O.A., K.K.K., J.A., S.Y.K. and K.U.E.O. All authors have read and agreed to the published version of the manuscript.

**Funding:** This research received no external funding.

**Institutional Review Board Statement:** Not applicable.

**Informed Consent Statement:** Not applicable.

**Data Availability Statement:** Data sharing is not applicable to this article as no new data were created or analyzed in this study.

**Conflicts of Interest:** The authors declare no conflict of interest.

## Appendix A

**Table A1.** Preferred Reporting Items for Systematic Reviews and Meta-Analyses extension for Scoping Reviews checklist [16].

| Section | Item | Prisma-ScR Checklist Item | Reported on Page |
|---|---|---|---|
| | | **TITLE** | |
| Title | 1 | Identify the report as a scoping review. | |
| Structured summary | 2 | Provide a structured summary that includes (as applicable): background, objectives, eligibility criteria, sources of evidence, charting methods, results, and conclusions that relate to the review questions and objectives. | |
| Rationale | 3 | Describe the rationale for the review in the context of what is already known. Explain why the review questions/objectives lend themselves to a scoping review approach. | |
| Objectives | 4 | Provide an explicit statement of the questions and objectives being addressed with reference to their key elements (e.g., population or participants, concepts, and context) or other relevant key elements used to conceptualize the review questions and/or objectives. | |
| Protocol and registration | 5 | Indicate whether a review protocol exists; state if and where it can be accessed (e.g., a Web address); and if available, provide registration information, including the registration number. | |
| Eligibility criteria | 6 | Specify characteristics of the sources of evidence used as eligibility criteria (e.g., years considered, language, and publication status), and provide a rationale. | |
| Information sources | 7 | Describe all information sources in the search (e.g., databases with dates of coverage and contact with authors to identify additional sources), as well as the date the most recent search was executed. | |
| Search | 8 | Present the full electronic search strategy for at least 1 database, including any limits used, such that it could be repeated. | |

**Table A1.** *Cont.*

| Section | Item | Prisma-ScR Checklist Item | Reported on Page |
|---|---|---|---|
| Selection of sources of evidence | 9 | State the process for selecting sources of evidence (i.e., screening and eligibility) included in the scoping review. | |
| Data charting process | 10 | Describe the methods of charting data from the included sources of evidence (e.g., calibrated forms or forms that have been tested by the team before their use, and whether data charting was done independently or in duplicate) and any processes for obtaining and confirming data from investigators. | |
| Data items | 11 | List and define all variables for which data were sought and any assumptions and simplifications made. | |
| Critical appraisal of individual sources of evidence | 12 | If done, provide a rationale for conducting a critical appraisal of included sources of evidence; describe the methods used and how this information was used in any data synthesis (if appropriate). | |
| Synthesis of results | 13 | Describe the methods of handling and summarizing the data that were charted. | |
| **RESULTS** | | | |
| Selection of sources of evidence | 14 | Give numbers of sources of evidence screened, assessed for eligibility, and included in the review, with reasons for exclusions at each stage, ideally using a flow diagram. | |
| Characteristics of sources of evidence | 15 | For each source of evidence, present characteristics for which data were charted and provide the citations. | |
| Critical appraisal within sources of evidence | 16 | If done, present data on critical appraisal of included sources of evidence (see item 12). | |
| Results of individual sources of evidence | 17 | For each included source of evidence, present the relevant data that were charted that relate to the review questions and objectives. | |
| Synthesis of results | 18 | Summarize and/or present the charting results as they relate to the review questions and objectives. | |
| **DISCUSSION** | | | |
| Summary of evidence | 19 | Summarize the main results (including an overview of concepts, themes, and types of evidence available), link to the review questions and objectives, and consider the relevance to key groups. | |
| Limitations | 20 | Discuss the limitations of the scoping review process. | |
| Conclusions | 21 | Provide a general interpretation of the results with respect to the review questions and objectives, as well as potential implications and/or next steps. | |
| **FUNDING** | | | |
| Funding | 22 | Describe sources of funding for the included sources of evidence, as well as sources of funding for the scoping review. Describe the role of the funders of the scoping review. | |

**Table A2.** Assessment of Multiple Systematic Reviews (AMSTAR-2) tool [17].

| | |
|---|---|
| **1. Did the research questions and inclusion criteria for the review include the components of PICO?** | |

| For Yes: | | |
|---|---|---|
| ☐ Population<br>☐ Intervention<br>☐ Comparator group<br>☐ Outcome | Optional (recommended)<br>☐ Timeframe for follow-up | ☐ Yes<br>☐ No |

| | |
|---|---|
| **2. Did the report of the review contain an explicit statement that the review methods were established prior to the conduct of the review and did the report justify any significant deviations from the protocol?** | |

| For Partial Yes:<br>The authors state that they had a written protocol or guide that included ALL the following:<br>☐ Review question(s)<br>☐ A search strategy<br>☐ Inclusion/exclusion criteria<br>☐ A risk of bias assessment | For Yes:<br>As for partial yes, plus the protocol should be registered and should also have specified:<br>☐ A meta-analysis/synthesis plan, if appropriate, *and*<br>☐ A plan for investigating causes of heterogeneity<br>☐ Justification for any deviations from the protocol | ☐ Yes<br>☐ Partial Yes<br>☐ No |
|---|---|---|

| | |
|---|---|
| **3. Did the review authors explain their selection of the study designs for inclusion in the review?** | |

| For Yes, the review should satisfy ONE of the following: | |
|---|---|
| ☐ *Explanation for* including only RCTs<br>☐ OR *Explanation for* including only NRSI<br>☐ OR *Explanation for* including both RCTs and NRSI | ☐ Yes<br>☐ No |

| | |
|---|---|
| **4. Did the review authors use a comprehensive literature search strategy?** | |

| For Partial Yes (all the following):<br>☐ Searched at least 2 databases (relevant to research question)<br>☐ Provided key word and/or search strategy<br>☐ Justified publication restrictions (e.g., language) | For Yes, should also have (all the following):<br>☐ Searched the reference lists/bibliographies of included studies<br>☐ Searched trial/study registries<br>☐ Included/consulted content experts in the field<br>☐ Where relevant, searched for grey literature<br>☐ Conducted search within 24 months of completion of the review | ☐ Yes<br>☐ Partial Yes<br>☐ No |
|---|---|---|

| | |
|---|---|
| **5. Did the review authors perform study selection in duplicate?** | |

| For Yes, either ONE of the following: | |
|---|---|
| ☐ At least two reviewers independently agreed on selection of eligible studies and achieved consensus on which studies to include<br>☐ OR two reviewers selected a sample of eligible studies <u>and</u> achieved good agreement (at least 80 percent), with the remainder selected by one reviewer. | ☐ Yes<br>☐ No |

| | |
|---|---|
| **6. Did the review authors perform data extraction in duplicate?** | |

| For Yes, either ONE of the following: | |
|---|---|
| ☐ At least two reviewers achieved consensus on which data to extract from included studies<br>☐ OR two reviewers extracted data from a sample of eligible studies <u>and</u> achieved good agreement (at least 80 percent), with the remainder extracted by one reviewer. | ☐ Yes<br>☐ No |

**Table A2.** *Cont.*

| 7. | Did the review authors provide a list of excluded studies and justify the exclusions? | | |
|---|---|---|---|

| For Partial Yes:<br>☐ Provided a list of all potentially relevant studies that were read in full-text form but excluded from the review | For Yes, must also have:<br>☐ Justified the exclusion from the review of each potentially relevant study | ☐ Yes<br>☐ Partial Yes<br>☐ No |
|---|---|---|

| 8. | Did the review authors describe the included studies in adequate detail? |
|---|---|

| For Partial Yes (ALL the following):<br>☐ Described populations<br>☐ Described interventions<br>☐ Described comparators<br>☐ Described outcomes<br>☐ Described research designs | For Yes, should also have ALL the following:<br>☐ Described population in detail<br>☐ Described intervention in detail (including doses where relevant)<br>☐ Described comparator in detail (including doses where relevant)<br>☐ Described study's setting<br>☐ Timeframe for follow-up | ☐ Yes<br>☐ Partial Yes<br>☐ No |
|---|---|---|

| 9. | Did the review authors use a satisfactory technique for assessing the risk of bias (RoB) in individual studies that were included in the review? |
|---|---|

| **RCTs**<br>For Partial Yes, must have assessed RoB from<br>☐ Unconcealed allocation, *and*<br>☐ Lack of blinding of patients and assessors when assessing outcomes (unnecessary for objective outcomes such as all-cause mortality) | For Yes, must also have assessed RoB from:<br>☐ Allocation sequence that was not truly random, *and*<br>☐ Selection of the reported result from among multiple measurements or analyses of a specified outcome | ☐ Yes<br>☐ Partial Yes<br>☐ No<br>☐ Includes only NRSI |
|---|---|---|
| **NRSI**<br>For Partial Yes, must have assessed RoB:<br>☐ From confounding, *and*<br>☐ From selection bias | For Yes, must also have assessed RoB:<br>☐ Methods used to ascertain exposures and outcomes, *and*<br>☐ Selection of the reported result from among multiple measurements or analyses of a specified outcome | ☐ Yes<br>☐ Partial Yes<br>☐ No<br>☐ Includes only RCTs |

| 10. | Did the review authors report on the sources of funding for the studies included in the review? |
|---|---|

| For Yes<br>☐ Must have reported on the sources of funding for individual studies included in the review. Note: Reporting that the reviewers looked for this information but it was not reported by study authors also qualifies | ☐ Yes<br>☐ No |
|---|---|

| 11. | If meta-analysis was performed did the review authors use appropriate methods for statistical combination of results? |
|---|---|

| **RCTs**<br>For Yes:<br>☐ The authors justified combining the data in a meta-analysis<br>☐ AND they used an appropriate weighted technique to combine study results and adjusted for heterogeneity if present.<br>☐ AND investigated the causes of any heterogeneity | ☐ Yes<br>☐ No<br>☐ No meta-analysis conducted |
|---|---|

**Table A2.** *Cont.*

| For NRSI | |
|---|---|
| For Yes:<br><br>☐ The authors justified combining the data in a meta-analysis<br>☐ AND they used an appropriate weighted technique to combine study results, adjusting for heterogeneity if present<br>☐ AND they statistically combined effect estimates from NRSI that were adjusted for confounding, rather than combining raw data, or justified combining raw data when adjusted effect estimates were not available<br>☐ AND they reported separate summary estimates for RCTs and NRSI separately when both were included in the review | ☐ Yes<br>☐ No<br>☐ No meta-analysis conducted |
| **12.** **If meta-analysis was performed, did the review authors assess the potential impact of RoB in individual studies on the results of the meta-analysis or other evidence synthesis?** | |
| For Yes:<br><br>☐ Included only low risk of bias RCTs<br>☐ OR, if the pooled estimate was based on RCTs and/or NRSI at variable RoB, the authors performed analyses to investigate possible impact of RoB on summary estimates of effect. | ☐ Yes<br>☐ No<br>☐ No meta-analysis conducted |
| **13.** **Did the review authors account for RoB in individual studies when interpreting/discussing the results of the review?** | |
| For Yes:<br><br>☐ Included only low risk of bias RCTs<br>☐ OR, if RCTs with moderate or high RoB, or NRSI were included the review provided a discussion of the likely impact of RoB on the results | ☐ Yes<br>☐ No |
| **14.** **Did the review authors provide a satisfactory explanation for, and discussion of, any heterogeneity observed in the results of the review?** | |
| For Yes:<br><br>☐ There was no significant heterogeneity in the results<br>☐ OR if heterogeneity was present the authors performed an investigation of sources of any heterogeneity in the results and discussed the impact of this on the results of the review | ☐ Yes<br>☐ No |
| **15.** **If they performed quantitative synthesis did the review authors carry out an adequate investigation of publication bias (small study bias) and discuss its likely impact on the results of the review?** | |
| For Yes:<br><br>☐ Performed graphical or statistical tests for publication bias and discussed the likelihood and magnitude of impact of publication bias | ☐ Yes<br>☐ No<br>☐ No meta-analysis conducted |
| **16.** **Did the review authors report any potential sources of conflict of interest, including any funding they received for conducting the review?** | |
| For Yes:<br><br>☐ The authors reported no competing interests OR<br>☐ The authors described their funding sources and how they managed potential conflicts of interest | ☐ Yes<br>☐ No |

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
