# Peer review of "Status of Omics Research Capacity on Oral Cancer in Africa: A Systematic Scoping Review Protocol"

_biomedinformatics, doi:10.3390/biomedinformatics3020022_

Round 1

Reviewer 1 Report

The paper deals with the interesting task of oral cancer in Africa capacity research.

It has a logical structure. But I feel some luck with the novelty and scientific sound of the paper because right now it looks more like a review or short summary of an existing dataset. The experimental section is not clear.

Suggestions:

1. Abstract should be extended by the obtained results, and proposed solution benefits and limitations.

2. Authors should provide a link to an open-access repository with the linguistic dataset used for training and performance evolution.

3. Methods section should be extended by the obtained results and structure in step by step model.

4. The paper Result and Discussion chapter numbers are the same.  

5. The conclusion section should be extended with approach limitations and prospects for future research.

Author Response

REVIEWER 1

The paper deals with the interesting task of oral cancer in Africa capacity research.

It has a logical structure. But I feel some luck with the novelty and scientific sound of the paper because right now it looks more like a review or short summary of an existing dataset. The experimental section is not clear.

Suggestions:

  1. Abstract should be extended by the obtained results, and proposed solution benefits and limitations.

  1. Authors should provide a link to an open-access repository with the linguistic dataset used for training and performance evolution.

  1. Methods section should be extended by the obtained results and structure in step by step model.

  1. The paper Result and Discussion chapter numbers are the same. 

  1. The conclusion section should be extended with approach limitations and prospects for future research.

RE: Thank you for the review comments. Our work is a study protocol. The recommendations you gave were not consistent with the structure of a scoping review protocol, unfortunately. To abide by international standards for reporting scoping review protocols, we will not be able to implement your recommendations. However, we appreciate your comments.

Reviewer 2 Report

Allow me to thank you for granting me both the opportunity and the privilege to read and review this article.

After carefully reading the manuscript “Status of Omics research capacity on Oral Cancer in Africa - a systematic scoping review protocolit is my recommendation that it should be “Rejected”.

The entire manuscript is nothing but a project. 

By phrasing the abstract in the future tense, I first thought that this seemed more like the abstract of a research project yet to be developed, rather than the abstract of a study that has been conducted. Apparently, it is so.

There are neither any results from the scoping review, nor any discussion has been presented. To publish this would be the same thing as turning the "Material and Methods" section of a scientific manuscript into an individual manuscript.

Moreover, I advise the authors to consider some existing literature on the topic (e.g.  Adeola HA et al. Cancer Cell Int. 2017; 17: 61).

Overall, the manuscript is poorly structured (to say the least) and adds nothing new to the existing body of literature.

Author Response

Allow me to thank you for granting me both the opportunity and the privilege to read and review this article.

After carefully reading the manuscript “Status of Omics research capacity on Oral Cancer in Africa - a systematic scoping review protocol”, it is my recommendation that it should be “Rejected”.

The entire manuscript is nothing but a project. 

By phrasing the abstract in the future tense, I first thought that this seemed more like the abstract of a research project yet to be developed, rather than the abstract of a study that has been conducted. Apparently, it is so.

There are neither any results from the scoping review, nor any discussion has been presented. To publish this would be the same thing as turning the "Material and Methods" section of a scientific manuscript into an individual manuscript.

Moreover, I advise the authors to consider some existing literature on the topic (e.g.  Adeola HA et al. Cancer Cell Int. 2017; 17: 61).

Overall, the manuscript is poorly structured (to say the least) and adds nothing new to the existing body of literature.

RE: Thank you for the review comments. Our work is a study protocol. The recommendations you gave were not consistent with the structure of a scoping review protocol, unfortunately. To abide by international standards for reporting scoping review protocols, we will not be able to implement your recommendations. However, we appreciate your comments.

Reviewer 3 Report

Was the review registered in prospero?

The discussion section is absent.

Author Response

Was the review registered in prospero?

RE: Thank you for the review comments. Scoping reviews are not registered in PROSPERO. PROSPERO registers systematic reviews only.

The discussion section is absent.

RE: Thank you for the review comments. This is a protocol. The inclusion of a discussion section is not a common practice. Additionally, the word count of our paper is now too/adequately large for a protocol. Hence, we are of the opinion that the conclusion section is sufficient. Thank you.

Round 2

Reviewer 1 Report

same comments, as the authors didn't take into account my previous comments at all, from my point of view, links and clear specification of the results is important for any kind of research.

Author Response

REVIEWER 1

same comments, as the authors didn't take into account my previous comments at all, from my point of view, links and clear specification of the results is important for any kind of research.

RE: Thank you very much for the comment. However, we are afraid that your concern about the inclusion of results in a scoping review protocol is not consistent with the international standards for such article types. We would like to refer to some scoping review protocols published in MDPI sister journals to clear your doubts on that:

Kunow, C.; Langer, B. Using the Simulated Patient Methodology in the Form of Mystery Calls in Community Pharmacy Practice Research: A Scoping Review Protocol. Pharmacy 2023, 11, 47. https://doi.org/10.3390/pharmacy11020047

Nunes, K.Z.; Grassi, J.; Lopes, A.B.; Rezende, L.D.A.; Cavalcanti, J.A.; Gomes, K.N.; Silva, J.A.D.d.; Lopes-Júnior, L.C. Clinical Indicators of Cardiovascular Risk in Adult Patients Undergoing Chemotherapy: A Protocol for Scoping Review. Pharmacoepidemiology 2023, 2, 35-41. https://doi.org/10.3390/pharma2010004

Lopes, R.H.; Silva, C.R.D.V.; Salvador, P.T.C.d.O.; Silva, Í.d.S.; Heller, L.; Uchôa, S.A.d.C. Surveillance of Drinking Water Quality Worldwide: Scoping Review Protocol. Int. J. Environ. Res. Public Health 2022, 19, 8989. https://doi.org/10.3390/ijerph19158989

We hope that the above articles have clarified your concerns. Thank you.

Reviewer 2 Report

The Introduction has been significantly improved and proper context has now been provided.

The data extraction and presentation has also been significantly improved and potential deviations are also now being considered.

Author Response

REVIEWER 2

The Introduction has been significantly improved and proper context has now been provided.

The data extraction and presentation has also been significantly improved and potential deviations are also now being considered.

RE: Thank you very much.

Reviewer 3 Report

Congratulations

Author Response

REVIEWER 3

Congratulations

RE: Thank you very much.